# The Anionic Polymerization of a *tert*-Butyl-Carboxylate-Activated Aziridine

**DOI:** 10.3390/polym14163253

**Published:** 2022-08-10

**Authors:** Chandan Giri, Jen-Yu Chang, Pierre Canisius Mbarushimana, Paul A. Rupar

**Affiliations:** Department of Chemistry and Biochemistry, The University of Alabama, Tuscaloosa, AL 35487-0336, USA

**Keywords:** polymer synthesis, aziridine, anionic polymerization, polyethyleneimine, *tert*-butyloxycarbonyl protecting group

## Abstract

*N*-Sulfonyl-activated aziridines are known to undergo anionic-ring-opening polymerizations (AROP) to form polysulfonyllaziridines. However, the post-polymerization deprotection of the sulfonyl groups from polysulfonyllaziridines remains challenging. In this report, the polymerization of *tert*-butyl aziridine-1-carboxylate (**BocAz**) is reported. **BocAz** has an electron-withdrawing tert-butyloxycarbonyl (BOC) group on the aziridine nitrogen. The BOC group activates the aziridine for AROP and allows the synthesis of low-molecular-weight poly(**BocAz**) chains. A ^13^C NMR spectroscopic analysis of poly(**BocAz**) suggested that the polymer is linear. The attainable molecular weight of poly(**BocAz**) is limited by the poor solubility of poly(**BocAz**) in AROP-compatible solvents. The deprotection of poly(**BocAz**) using trifluoroacetic acid (TFA) cleanly produces linear polyethyleneimine. Overall, these results suggest that carbonyl groups, such as BOC, can play a larger role in the in the activation of aziridines in anionic polymerization and in the synthesis of polyimines.

## 1. Introduction

Polyethylenimine (PEI) is a polymer with a high amine density that is widely studied for a wide range of applications, including non-viral gene transfection [1,2,3,4,5], metal chelation [6,7,8], and CO_2_ capture [9,10,11,12,13]. PEI exists in two forms: branched (bPEI) and linear (lPEI). The bPEI type forms from the cationic ring-opening polymerization (CROP) of aziridine (Figure 1A) [14,15], while the lPEI form is typically prepared through the hydrolysis of poly(2-oxazolines), which itself is synthesized by the CROP of 2-oxazolines [16,17,18]. In particular, lPEI is often sought over bPEI due to its better reported biocompatibility and reduced cytotoxicity [19,20,21,22]. 

The anionic-ring-opening polymerization (AROP) of sulfonylaziridines [14], first reported by Toste and Bergman [23], provides a potential alternative route to linear polyethylenimine-like polymers (Figure 1B). The *N*-substituted sulfonyl groups prevent branching reactions during the polymerization by deactivating the lone pairs on the nitrogen atoms of the developing polymer chains. This approach has been further extended by Wurm [24,25,26,27,28,29], Taton [30,31], Carlotti [32,33], Rupar [34,35,36], Guo [37], Yoon [38], Zhang [39], and Hadjichristidis [40] for the polymerization of numerous polymers containing sulfonylaziridine (and sulfonylazetidine) [14,36,41,42,43]. 

A significant challenge in this field is the removal of the sulfonyl groups from poly(sulfonylaziridines) to form the corresponding polyimines. Strong reducing agents and harsh conditions are often required for the complete desulfonylation of poly(sulfonylaziridines) [35,44]. The development of novel *N*-activated aziridine monomers with more easily removable electron-withdrawing groups remains highly desirable. We now report on the anionic polymerization of *tert*-butyl aziridine-1-carboxylate (**BocAz**). **BocAz** has a *tert*-butyloxycarbonyl (Boc)-activating group on the nitrogen atom of aziridine (Figure 1C). The Boc group was chosen because of its tolerance towards basic conditions and nucleophilic reagents [45,46,47]. Due to the Boc group’s electron-withdrawing properties, we propose that Boc will deactivate the lone pair of electrons on the nitrogen atoms of the growing polymer chain to prevent polymer branching. An important advantage of Boc compared to sulfonyl groups is that Boc can undergo deprotection using relatively mild acidic conditions. There are examples of the nucleophilic ring-opening of **BocAz** in the synthesis of small molecules, although its polymerization chemistry has not been explored [48,49,50]. 

## 2. Materials and Methods

Unless stated otherwise, all reagents and solvents were purchased from commercial suppliers and used directly without further purification. Solvents used in polymerizations (DMF, THF, toluene, DMSO) were purchased as anhydrous-grade and were stored over molecular sieves in a glove box prior to use. BuN(H)Ts were synthesized following procedures in the literature [51]. All manipulations were carried out under a nitrogen atmosphere. Nuclear magnetic resonance (^1^H NMR and ^13^C NMR) spectra were recorded with a Bruker (Bruker, Billerica, MA, USA) AVANCE NEO III HD 500 spectrometer equipped with a cryoprobe. MALDI-TOF mass spectra were obtained using a Bruker (Bruker, Billerica, MA, USA) Ultraflex I MALDI-TOF mass spectrometer equipped with a pulsed 50-hertz, 337-nanometer nitrogen laser. Gel permeation chromatography (GPC) was performed using a TOSOH (Tokyo, Japan) BIOSCIENCE Gel permeation chromatograph equipped with an RI detector, an automatic sampler, a pump, an injector, an inline degasser, a column oven (35 °C), two in-series TSKgel SuperAWM-H SEC columns, and a TSKgel SuperAW2500 column. HFIP with CF_3_COOK (3.0 mg/mL) was used as the mobile phase at a flow rate of 0.1 mL/min. 

Synthesis of *tert*-butyl (2-hydroxyethyl)carbamate

To ethanolamine (1.52 mL, 25.1 mmol) in THF (100 mL) was added di-*tert*-butyl dicarbonate (5.5 g, 25 mmol). The reaction mixture was stirred at 28 °C for 72 h. THF was then removed under vacuum and the residue was redissolved in dichloromethane (30 mL), washed with 1% HCl solution (30 mL), brine (2 × 30 mL), and deionized water (30 mL), then dried over anhydrous MgSO_4_ and filtered. Removal of solvent under vacuum yielded a yellow oil identified as *tert*-butyl (2-hydroxyethyl)carbamate (2.5 g, 70.0%). ^1^H NMR (360 MHz. CDCl_3_): 6.67 (t, J = 10.6 Hz, 1H), 4.57 (t, J = 11.35 Hz, 1H), 3.34 (q, J = 18.39 Hz, 2H), 2.98 (t, J = 18.62 Hz, 2H), 1.37 (s, 9H) (Appendix A). The spectra matched that reported in the literature [52].

Synthesis of **BocAz**

To tert-butyl (2-hydroxyethyl)carbamate (0.81 g, 5.0 mmol) in diethyl ether (40 mL) was added KOH (1.15 g, 20.5 mmol) and tosyl chloride (1.03 g, 5.40 mmol). The reaction mixture was stirred at room temperature for 72 h, filtered, and washed with deionized water. The ether was evaporated to yield a greenish yellow oil. Purification by silica chromatography and vacuum distillation over CaH_2_ yielded **BocAz** as a clear viscous liquid (0.31 g, 72%). ^1^H NMR (500 MHz, CDCl_3_): 2.10 (s, 4H), 1.42 (s, 9H) (Appendix A). ^13^C NMR (125 MHz, CDCl_3_): δ 162.78, 81.08, 27.86, 25.69. HRMS (EI): calcd for C_7_H_13_NO_2_ (M+) 144.1015, found 144.1019 (Appendix A).

Preparation of Initiator BuN(K)Ts in DMF

BuN(H)Ts (62 mg, 0.27 mmol) and KHMDS (54 mg, 0.27 mmol) were combined in 2 mL of anhydrous DMF. The reaction mixture was stirred for 1 h before use in a polymerization reaction. The solution was assumed to be 0.135 M BuN(K)Ts and was used without characterization. 

Example Procedure for the Polymerization of **BocAz**

To **BocAz** (30 mg, 0.21 mmol) in anhydrous DMF (0.5 mL) was added 77 μL of the 0.135 M BuN(K)Ts (0.0104 mmol) solution in DMF. This resulted in an monomer-to-initiator ratio of ca. 20:1. The reaction mixture was heated at 50 °C overnight. The resulting gel-like mixture was dispersed in MeOH (10 mL) to yield a white solid, which was collected by centrifugation. See Figure 1 and Table 1 for the ^1^H NMR spectra of poly(BOC) and for reaction yields. 

Example procedure for the synthesis of lPEI from Poly(**BocAz**)

Poly(**BocAz**) (30 mg) was dissolved in DCM (4 mL). TFA (1 mL) of TFA was added to the reaction mixture at 0 °C. The reaction mixture was stirred for overnight. The DCM was evaporated. The residue was washed with Et_2_O followed by 1(M) NaOH and water. The yield was 7.56 mg (85%). The ^13^C NMR showed a single signal at 47 ppm, which is characteristic of lPEI (Appendix A).

## 3. Results and Discussion

**BocAz** has been reported previously [48,49,50], but its polymerization has not been explored. We synthesized **BocAz** from ethanolamine by first reacting ethanolamine with di-*tert*-butyl dicarbonate to form *tert*-butyl (2-hydroxyethyl)carbamate (Figure 2A). The subsequent reaction with tosyl chloride in the presence of KOH produced the desired **BocAz** as a viscous liquid.

In prior reports on the anionic polymerization of sulfonylaziridines, a wide array of initiators and solvent systems have been used [14]. In our initial efforts, we used conditions similar to those reported by Bergman and Toste [23]. The polymerizations of the **BocAz** were performed in DMF at 50 °C using BuN(K)Ts as the initiators, with a monomer-to-initiator ratio of 20:1 (Figure 2B, Table 1). The BnN(K)Ts were generated in situ from BuN(H)Ts and KHMDS, and then added to a DMF solution of **BocAz**. A few hours after the addition of the BuN(K)Ts to the **BocAz** solution, the viscosity of the reaction mixture greatly increased, before finally forming a gel. The gel was dispersed into a methanol solution to produce a white powder, which was collected and dried. 

The ^1^H NMR spectrum of the white power in the CDCl_3_ was consistent with the expected structure of the poly(**BocAz**) (Figure 2B and Figure 1). The ^1^H NMR spectrum was dominated by resonances at 1.44 ppm and 3.30 ppm, which were attributed to the Boc *tert*-butyl group and polymer methylene protons, respectively (Figure 1). The integration ratio of these signals was close to the expected 9:4, although the accurate integration of the *tert*-butyl protons was complicated by proximity of the H_2_O resonance and likely contributions from the initiator’s methylene protons. Signals originating from the BuN(K)Ts initiator were visible in the ^1^H NMR spectrum, and NMR end-group analysis suggested that the average degree of polymerization of the poly(**BocAz**) was about 15.2 repeat units (*ca.* 2.4 kDa, including the initiator) (Figure 1). The ^13^C NMR spectrum of the poly(**BocAz**) was relatively simple, suggesting little to no branching (Appendix A).

The MALDI-TOF MS spectrum of the poly(**BocAz**) was dominated by a series of signals with the expected 143 *m*/*z* spacing, corresponding to the molar mass of the **BocAz** monomer. The signals from the main series exactly matched the poly(**BocAz**) chains initiated by the BuN(K)Ts, terminated by a proton, and ionized by a sodium ion (Appendix A). The GPC showed the presence of low-molecular-weight material, with a M_n_ of about 1.16 kDa (vs a PMMA standard), although this weight approached the lower molecular weight limit of the instrument.

Prior work on the polymerization of sulfonylaziridines demonstrated control over poly(sulfonylaziridine) molecular weights through varying the ratio of the monomer to the initiator [23]. To determine whether the polymerization of **BocAz** can also be controlled, we performed a series of polymerizations in DMF, in which the ratio of the **BocAz**:BuN(K)Ts varied between 20 and 80 (Table 1). In all cases, either precipitates formed during the course of the polymerizations or the reaction mixtures gelled. The resulting polymers were characterized by GPC and ^1^H NMR spectroscopy. Both the GPC and ^1^H NMR spectroscopic end-group analyses showed that the monomer-to-initiator ratio had some impact on the molecular weight of the resulting poly(**BocAz**). Unfortunately, the dispersity (Ð) was very high, and it was inconsistent with a controlled polymerization. We believe the high dispersity was due to the precipitation of the poly(**BocAz**) during the course of the reaction. The polymerizations performed with higher **BocAz**:BuN(K)Ts ratios did not produce high-molecular-weight material. 

We searched for conditions in which poly(**BocAz**) did not prematurely precipitate during the polymerization of the **BocAz**. However, attempts at polymerizing **BocAz** in DMSO, toluene, [P_6,6,6,14_][Tf_2_N] (a phosphonium ionic liquid) [53], or THF resulted in either no reaction or identical outcomes to the polymerizations in the DMF. Increasing the temperature to 80 °C in the DMF did not prevent the precipitation of the poly(**BocAz**). 

The poor solubility of poly(**BocAz**) mirrors that of some poly(sulfonylaziridines) [34,35]. In Bergman and Toste’s initial report on the AROP of sulfonylaziridines, they found that *N*-methanesulfonylaziridine (which lacks substitution at the 2-position) produced only short oligomers due to poor polymer solubility [23]. By contrast, 2-methyl *N*-methanesulfonylaziridine polymerized to a high molecular weight. The poor solubility of sulfonylaziridines that lack 2-substitution has been ascribed to strong interchain interactions and polymer crystallinity [35]. The contrasting solubility of polymers formed from 2-substituted sulfonylaziridines has been explained by their atacticity, which is thought to interfere with interchain packing. 

We propose that the poor solubility of poly(**BocAz**) occurs for similar reasons to that observed for the poly(sulfonylaziridines). Specifically, the absence of substitution at the 2-position in the poly(**BocAz**) backbone permits strong interchain packing, making the polymer insoluble in common aprotic solvents. In future work, we will explore the polymerization of 2-methyl-substituted **BocAz** derivatives. 

An alternative explanation for the poor solubility of poly(**BocAz**) is that poly(**BocAz**) is crosslinked. However, the fact that all the samples of poly(**BocAz**) were soluble in the DCM and CDCl_3_ suggests that crosslinking was not present. This also suggests that it may be possible to achieve high-molecular-weight poly(**BocAz**) via AROP if a suitable AROP-compatible solvent system can be identified. Unfortunately, we have been unable to identify such a solvent. 

We believe that the AROP of **BocAz** proceeds via a similar mechanism to the polymerization of *N*-sulfonylaziridines (Figure 3) [14]. Initiation occurs when BuN(K)Ts nucleophically attack the aziridine ring of **BocAz**, leading to ring opening and the formation of a new aza anion. Propagation occurs through successive nucleophilic attacks of the aza anions of the growing chain end with additional **BocAz** molecules. We suspect that polymer-chain termination via protonation occurs upon precipitation into methanol. 

Although it was not possible to synthesize high-molecular-weight poly(**BocAz**), poly(**BocAz**) could still prove to be valuable as a source for low-molecular-weight lPEI. We attempted to deprotect poly(**BocAz**) using trifluoroacetic acid (TFA) in a solution of DCM. The ^13^C spectra of the resulting white powder were fully consistent with the formation of lPEI (Appendix A). The ^13^C NMR spectrum of the white powder was especially convincing as it consisted of a single signal at 47 ppm, which is characteristic of lPEI [18]. By contrast, bPEI exhibits a much more complex ^13^C NMR spectrum [15].

## 4. Conclusions

In summary, we reported on the anionic ring-opening polymerization (AROP) of *tert*-butyl aziridine-1-carboxylate (**BocAz**). The resulting poly(**BocAz**) was linear but had poor solubility in most solvents. This poor solubility limits access to high-molecular-weight polymers as poly(**BocAz**) precipitates prematurely during polymerization. The Boc group of poly(**BocAz**) is easily removed using trifluoroacetic acid (TFA) to form low-molecular-weight linear polyethyleneimine (lPEI). 

Overall, our results show that the Boc group is a potential alternative to sulfonyl groups for activating aziridines for anionic polymerizations. It also suggests that carbonyl-activating groups could play a larger role in the growing field of aziridine-polymerization chemistry. 

## Data Availability

NMR, GPC, and MALDI-TOF MS data can be found at DOI 10.17605/OSF.IO/ZH46M.

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
