# Peer review of "The Anionic Polymerization of a *tert*-Butyl-Carboxylate-Activated Aziridine"

_polymers, 2022, doi:10.3390/polym14163253_

Round 1
Reviewer 1 Report
This work describes an anionic polymerization of tert-butyl aziridine-1-carboxylate (BocAz). The tert-butyloxycarbonyl (Boc) was used as the electron withdrawing group on the aziridine nitrogen, allowing for anionic ring-opening polymerizations. Low molecular weight linear poly(BocAz) was synthesized and deprotection of poly(BocAz) efficiently yielded linear polyethyleneimine. The biggest novelty is that this is the first report of anionic ring-opening polymerizations using carbonyl protected aziridine as monomer. Unfortunately, the dispersity was high and the AROP is not controlled due to the poor solubility of poly(BocAz). As mentioned by the authors, a combination of solvent for living AROP can’t be identified. I believe this is an useful study for the development of AROP based on non-sulfonyl-aziridine monomers as an alternative synthesis of polyethylenimine. I recommend accepting this work to Polymers. I still encourage the authors to screening combination of solvents to either make the AROP living or achieve high DP & Mw regardless of livingness of polymerization.
Author Response
We thank the reviewer for their kind comments. As the reviewer suggests for future work, we will continue to find conditions to achieve high molecular weight polymer.
Reviewer 2 Report
This is a well written report of sound experimental synthetic polymer chemistry, which deserves publication. As a minor comment, I would like to mention that the certain aspects of the paper (polymerization mechanism, reason for low molar mass) are rather speculative but this is recognized by the authors.
Author Response
We than thank the reviewer for their comments.